

# Electrochemical sensors onboard a Zeppelin NT: In-flight evaluation of low-cost trace gas measurements

Tobias Schuldt[1], Georgios I. Gkatzelis[1], Christian Wesolek[1], Franz Rohrer[1], Benjamin Winter[1], Thomas A. J. Kuhlbusch[2], Astrid Kiendler-Scharr[1], Ralf Tillmann[1]

[1]Institute of Energy and Climate Research, IEK-8: Troposphere, Forschungszentrum Jülich GmbH, Jülich, Germany
[2]Federal Institute for Occupational Safety and Health (BAuA), Dortmund, Germany

*Correspondence to*: Tobias Schuldt (t.schuldt@fz-juelich.de), Georgios I. Gkatzelis (g.gkatzelis@fz-juelich.de)

## Abstract

In this work we used a Zeppelin NT equipped with six sensor setups, each composed of four different low-cost electrochemical sensors (ECS) to measure nitrogen oxides (NO and $NO_2$), carbon monoxide, and Ox ($NO_2 + O_3$) in Germany. Additionally, a MIRO MGA laser absorption spectrometer was installed as a reference device for in-flight evaluation of the ECS. We report the influence of temperature on the NO and $NO_2$ sensor outputs, but also find a shorter time scale (1 s) dependence of the sensors on the relative humidity gradient. To account for these dependencies, we developed a correction method that is independent of the reference instrument. After applying this correction to all individual sensors, we compare the sensor setups with each other and to the reference device. For the intercomparison of all six setups we find good agreements with $R^2 \geq 0.8$, but different precisions for each sensor in the range of 1.45 to 6.32 ppb. The comparison to the reference device results in an $R^2$ of 0.88 and a slope of 0.92 for $NO_x$ (NO + $NO_2$). Furthermore, the average noise (1 σ) of the NO and $NO_2$ sensors reduces significantly from 6.25 ppb and 7.1 ppb to 1.95 ppb and 3.32 ppb, respectively. Finally, we highlight the potential use of ECS in airborne applications, by identifying different pollution sources related to industrial and traffic emissions during multiple commercial and targeted Zeppelin flights in spring 2020. These results are a first milestone towards the quality-assured use of low-cost sensors in airborne settings without a reference device, e.g., on unmanned aerial vehicles (UAVs).

## Keywords

Sensor, low cost, airborne, air pollution, trace gas

## 1 Introduction

The effects of poor air quality are manifold – they are environmental (e.g., Mclaughlin, 1985; Bytnerowicz et al., 2007), economic (e.g., Quah and Boon, 2003), and health-related (e.g., Kampa and Castanas, 2008; Von Schneidemesser et al., 2020). Ambient air quality is dependent on the level of pollutant concentrations in the lower troposphere that includes both airborne particles and gaseous substances. Increasing concentrations of target pollutants, i.e., particulate matter, PM, nitrogen dioxide ($NO_2$), and ozone ($O_3$) increase the risk of cardiovascular, respiratory, and cerebrovascular mortality for short-term (Orellano





et al., 2020) as well as long-term exposure (Huangfu and Atkinson, 2020; Chen and Hoek, 2020). Furthermore, these pollutants can influence climate change with either cooling or warming effects (IPCC, 2021). Historically, in order to limit such impacts the World Health Organization (WHO, 2021) and the Intergovernmental Panel for Climate Change (IPCC, 2021) set global guidelines for countries to achieve. Quantification of pollutants is, therefore, an essential procedure for assessing air quality

and climate change and the first step through which subsequent action can be taken.

In Europe, this has been largely achieved via ground-based monitoring networks, such as the European Environment Agency's European Monitoring and Evaluation Programme (EMEP; http://ebas.nilu.no), or even infrastructures such as the Aerosols, Clouds, and Trace Gases Research Infrastructure (ACTRIS; https://www.actris.eu), that provide high-quality data for criteria pollutant concentrations around the world. The EU Air Quality Directive 2008/50/EC with its amendment 2015/1480/EC was

introduced to create uniform requirements for air quality measurements. In Germany, the measuring stations are operated by the State Environmental Agencies and the Federal Environment Agency in accordance with the Air Quality Directive's specifications. However, such ground-based measurements are typically stationary and thus cover local air quality trends with limited insights on the vertical distribution of pollutants or small-scale spatial gradients (Apte et al., 2017; Messier et al., 2018). Furthermore, maintenance and operation of such networks at the spatial resolution needed to inform decision makers, may

exceed available budgets.

In the last years, development of *low-cost sensors* e.g., electrochemical sensors (ECS), whose costs are two to three orders of magnitude lower than those of typical laboratory-grade devices, have been used as an alternative and affordable option to perform measurements that cover multiple locations (Popoola et al., 2018; Rai et al., 2017; Shusterman et al., 2016; Sun et al., 2016; Mead et al., 2013). Such sensors have been extensively used and evaluated alongside measuring stations (Popoola et al.,

2018; Sahu et al., 2021; Mead et al., 2013; Dallo et al., 2021; Spinelle et al., 2017, 2015) and have recently been extended for airborne applications (Villa et al., 2016; Schuyler and Guzman, 2017; Gu et al., 2018; Mawrence et al., 2020; Pochwala et al., 2020; Pang et al., 2021; Bretschneider et al., 2022). ECSs are light, compact in size, and of low power consumption (Alphasense, 2019b, a, f, e, d) - properties required for use on unmanned aerial vehicles (UAV); however, evaluation compared to reference devices in such airborne applications still remains a challenge.

Data quality assurance of ECSs is an essential step due to their cross-sensitivities to a wide range of influencing factors. These include meteorological parameters such as temperature (Mead et al., 2013; Popoola et al., 2016) and relative humidity (Samad et al., 2020; Wei et al., 2018), as well as cross-sensitivities to other gases (Mueller et al., 2017; Pang et al., 2018; Lewis et al., 2016). Furthermore, the gradients of meteorological parameters like the rate of relative humidity changes (% $s^{-1}$) can influence the sensor signal with slow recovery times of up to hours (Mueller et al., 2017; Pang et al., 2018; Pang et al., 2017). There are

two possible scenarios to compensate for such interferences: hardware modifications and post-processing of the data. Examples for hardware modifications are the introduction of a fourth electrode in the typical three-electrode electrochemical sensor to compensate for zero shifts (Baron and Saffell, 2017) and the implementation of a filter for specific cross interfering gases, e. g. for $O_3$ in an $NO_2$ ECS (Hossain et al., 2016). On the post-processing side, a growing variety of methods are used to obtain sufficient data quality. These include parametric algorithms such as multiple linear regressions (Wei et al., 2018), but also non-





parametric methods such as decision trees (Zimmerman et al., 2018), artificial neural networks (Han et al., 2021) or other
numerical models (Cross et al., 2017). M*achine learning* is often used as an equivalent term for non-parametric models (WMO,
2018). It has the advantage to identify more complex interferences in large datasets, such as non-linearities, time dependencies,
or combined interferences. A limitation, however, is that the cause of these interferences remains unknown when using
machine learning models, in contrast to e.g., linear regression methods where such dependencies can be identified.

In this work, we evaluate the performance of low-cost sensors and develop a correction method that accounts for ECS
interferences based on real-time airborne observations onboard a Zeppelin NT. This includes the measurements of CO, NO,
$NO_2$ and Ox ($NO_2 + O_3$) that are compared to the MIRO MGA (Tillmann et al., 2022) used as a reference device to evaluate
the performance of the sensors. We show that ECSs can be used for reliable, in situ trace gas measurements and highlight their
potential for airborne applications aboard UAVs.

## 2 Methods

### 2.1 Experimental setup / platform

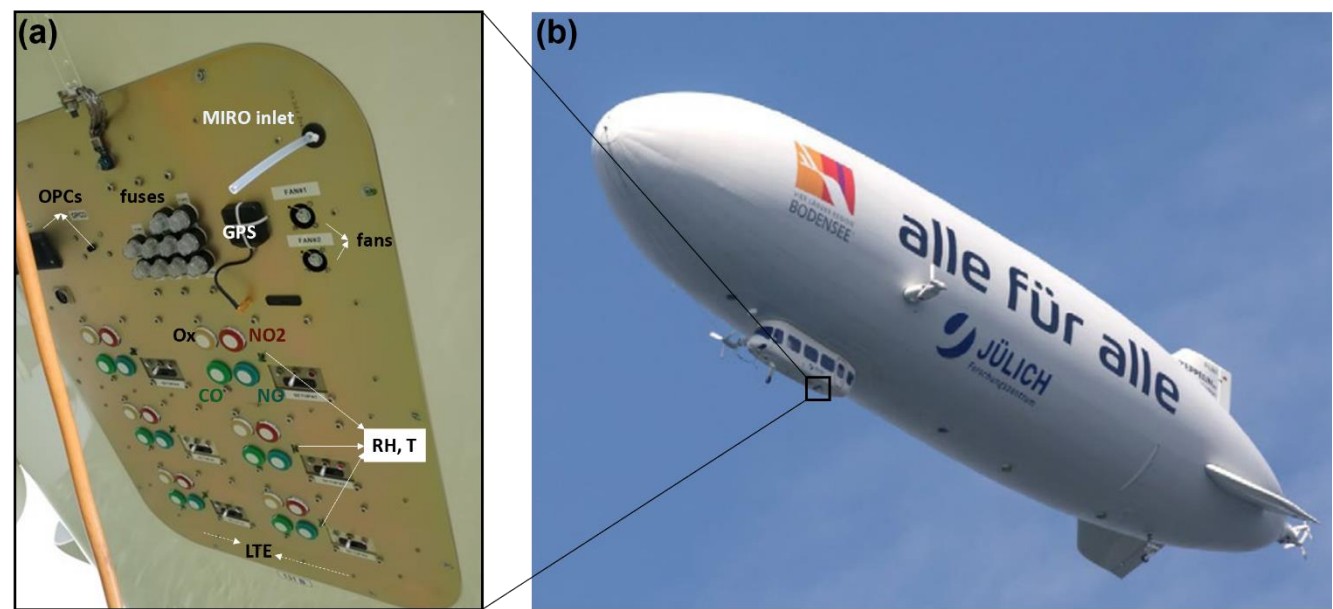

**Figure 1: (a) Hatch box including the sensor setups located on the bottom of the (b) Zeppelin NT (© Forschungszentrum Jülich /
Ralf-Uwe Limbach) that was used as the measurement platform in this work. The reference device, MIRO MGA, was installed
inside the gondola, but with the inlet line right beside the sensor setups, shown in the top right of panel (a).**

Figure 1 shows the experimental setup for the in situ airborne measurements. This includes a Zeppelin NT (b) as a measurement
platform equipped with a hatch box (a), located on the bottom of the airship. The Zeppelin NT is particularly suitable for
planetary boundary layer (PBL) measurements due to its long flight duration (up to 20 h) at low altitudes below 1 km, a high



payload of around 1000 kg, and good manoeuvrability. This allows measurements within the PBL to investigate the influence of different urban emission sources on air quality (Tillmann et al., 2022).

Six electrochemical sensor (ECS) setups are installed at the hatch box, together with a GPS module (Adafruit Ultimate GPS), Long Term Evolution (LTE) equipment for remote access, ventilation fans to regulate the hatch box pressure, two optical particle counters, and fuses to protect the electronics. The sensor inlets are located outside of the hatch box and exposed to ambient air via diffusion. A sensor setup, as shown in Fig S1, consists of four ECS to measure the trace gases CO, NO, $NO_2$

and Ox ($NO_2 + O_3$), a Telaire ChipCap 2 sensor for temperature and relative humidity measurements and a self-developed printed circuit board (PCB) for managing and saving the incoming data with a frequency of 1 Hz. Further specifications of the sensors are given in Table 1. The setups are powered from the Zeppelin NT and further supported by two batteries to provide uninterrupted power to the sensors.

**Table 1: Specifications of each sensor setup.**

| Parameter | Sensor | Principle | Response time[a] | Accuracy[a] | Range[a] |
|---|---|---|---|---|---|
| CO | Alphasense CO-B4 | Amperometric | < 30 s ($t_{90}$) | ±4 ppb (precision, 2σ) | 0-1000 ppm |
| NO | Alphasense NO-B4 | Amperometric | < 45 s ($t_{90}$) | ±15 ppb (precision, 2σ) | 0-20 ppm |
| $NO_2$ | Alphasense NO2-B43F | Amperometric | < 80 s ($t_{90}$) | ±15 ppb (precision, 2σ) | 0-20 ppm |
| Ox ($NO_2 + O_3$) | Alphasense Ox-B431 | Amperometric | < 80 s ($t_{90}$) | ±15 ppb (precision, 2σ) | 0-20 ppm |
| Aerosol (Size distribution, $PM_1$, $PM_{2.5}$, $PM_{10}$)[b] | Alphasense OPC-N3 | Light scattering | 1-30 s (sampling interval) | - | 0.35 - 40 µm |
| Temperature | Telaire ChipCap 2 | CMOS | 5 s ($t_{63}$) | ±0.3 °C | -40 to 125 °C |
| Relative Humidity | | Capacitive | 4 s ($t_{63}$) | ±2.0 % RH (20-80 % RH) | 0 - 100 % RH (Non-Condensing) |

[a] Specifications given in the corresponding data sheets.
[b] Connected to setup #1 and #4 but not used in this work.

In this study we focus on the measurements of NO and $NO_2$ ($NO_x$) using electrochemical sensors from Alphasense (UK,
Essex). Laboratory evaluation and optimization of the CO and $O_x$ sensor performance is currently ongoing, and the focus of a future study as further discussed in section 2.3.2. Here, the $NO_x$ measurements are performed using amperometric gas sensors that have four electrodes: a *working electrode* (*WE*), a *counter electrode* (*CE*), a *reference electrode* (*RE*), and the *auxiliary electrode* (*AUX*). The measuring principle is based on a redox reaction that takes place at the *WE* and the *CE* (reduction, oxidation). The *RE* keeps the *WE* at a constant potential to force the desired electrochemical reaction of the analyte on the
three-phase boundary (electrode, electrolyte, gas). The resulting charges are transferred to each electrode in the form of ions via the electrolyte solution and in the form of electrons via an external circuit. The resulting current of the electron transfer is





the measurement signal of the sensor. This current is exactly proportional to the concentration of the analyte when the sensor is operated under appropriate diffusion-limited conditions. Many kinetic factors, such as the mass transfer of the analyte to the electrode as well as the electrocatalytic activity of the electrode material, can be adjusted via the design of the sensor (Stetter

and Li, 2008). *AUX* has the same design as the *WE*, but is fully immersed in the electrolyte and has no interface with the gas phase (Baron and Saffell, 2017). Therefore, the *AUX* signal can be used to correct the *WE* signal from influences on the *WE* other than the analyte (e.g., temperature). The measured *WE* and *AUX* currents are converted into a voltage signal using an individual sensor board (ISB, Alphasense). Finally, this signal is digitised by the measurement board and then recorded on an SD card in binary format and can also be transferred serially.

Besides the measurement equipment in the hatch box, a MIRO MGA10-GP multi-compound gas analyzer was installed in the gondola. The MIRO MGA measures the amount fractions of ten trace gases (NO, $NO_2$, $O_3$, $SO_2$, CO, $CO_2$, $CH_4$, $H_2O$, $NH_3$, $N_2O$) by direct laser absorption spectroscopy. In this study it is used as a reference system for the ECS and enables a direct performance evaluation of the ECS in an airborne setting. Hundt et al. (2018) and Liu et al. (2018) provide more in-depth details about the MIRO, while more information on its use on the Zeppelin NT are given by Tillmann et al. (2022). For an

accurate intercomparison to the ECSs, a perfluoroalkoxy alkane (PFA) inlet line with a length of 8 m was placed next to the ECSs in the hatch box (Fig. 1 (a)) and connected to the MIRO. The volumetric flowrate used for the MIRO was 1.2 SLM resulting in a residence time of the gas inside the line of around 5 s.

## 2.2 Location and dates

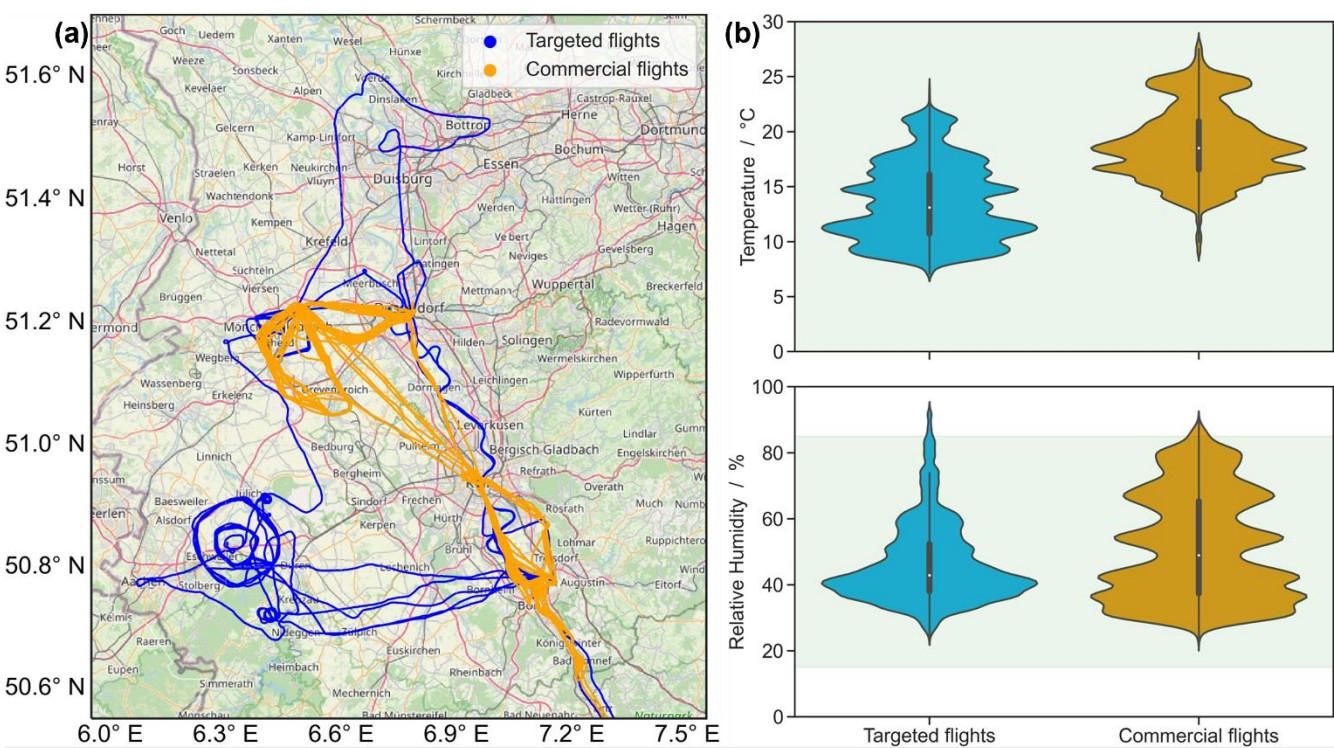

**Figure 2: (a) Map with flight paths in North Rhine-Westphalia, Germany (© OpenStreetMap contributors 2021. Distributed under the Open Data Commons Open Database License (ODbL) v1.0), and (b) frequency distribution of the corresponding temperature and relative humidity in-flight values for the targeted (blue) and commercial (orange) flights - illustrated in violin plots. The manufacturer-specified limits within which the sensors should be used are depicted in shaded green in the background.**

Figure 2 (a) depicts a map with the flight paths for our measurements in 2020. The flights took place within two periods in mid to late spring, with targeted research flights performed from 29 April 2020 to 09 May 2020 and measurements during commercial flights from 27 May 2020 to 15 June 2020. All flights were over Germany, and predominantly over North Rhine-Westphalia except for the transfer flights on 29 April 2020 and 27 May 2020 from Friedrichshafen to North Rhine-Westphalia, and from North Rhine-Westphalia back to Friedrichshafen on 09 May 2020 and 15 June 2020. During the targeted flights, specific emission sources e.g., a power plant were targeted as well as cities and rural areas. More detailed information on individual flights is provided by Tillmann et al. (2022). Figure 2 (b) shows the in-flight measured temperature and relative humidity values. According to the manufacturers specifications the sensors should be used in the range from -30 °C to 40 °C and 15 % to 85 % relative humidity (Alphasense, 2019b, a, e, d). Nearly the entire data set is within these specifications with 1 % and 99 % percentiles at 8.4 °C (1 %), 25.7 °C (99 %) and 28.0 % RH (1 %), 84.0 % RH (99 %), respectively. Furthermore, the manufacturer recommendations are given for continuous exposure at high or low RH which was not observed for the whole dataset hence limiting the influence of such interferences for this study.



### 2.3 Data processing

The quality of sensor data can be influenced by various parameters including transmission errors to the measurement board and electromagnetic interference between the devices (Alphasense, 2013), as well as defective sensor components. In this
work, we use multiple ECS sensors in parallel to track defective sensors after following the steps that are outlined below.

### 2.3.1 Time synchronization and noise reduction

The clocks on each of the six sensor setups were manually pre-set, therefore time synchronization was not ensured. In the first step we chose a *master setup*, here setup #2 that was operational throughout all flights with a data coverage of 99 %. Next, a time shift was applied to the other setups to match it. To find the optimal values for this time shift, setup X was shifted from -
60 s to +60 s stepwise by 1 s, performing a linear regression analysis (setup X(t+-x) vs. setup #2). The linear regression resulting in the highest coefficient of determination, $R^2$, was used to correct for the time difference of each setup to the master setup. This was done using the full dataset of each period of flights, targeted and commercial, resulting in shifts within ±15 s and an average time drift of ±0.31 s week$^{-1}$, leading to a maximum drift of $< \pm 1$ s during each single period of flights.

In the next step the data of the sensors and the MIRO MGA whose clock is set via LTE were synchronized. This
synchronization step made it possible to properly compare the sensors with each other and with the reference instrument. Lastly, to reduce the noise of the ECS signals, a Savitzky-Golay filter (Savitzky and Golay, 1964) was used. Here, a polynomial regression with a window size of 2 n + 1 adjacent data points is solved by linear least squares. A window of 11 s (n = 5) and a polynomial degree of 3 was found to be optimal to smooth the signals without altering the analyte peaks. In addition, meteorological measurements including temperature and relative humidity were evaluated by performing linear regressions
between the six ChipCap 2 sensors as shown in Fig. S2. From this correlation analysis it is evident that the sensors of setups #4, #6 and partly #5 provide erroneous data. We therefore calculate and use throughout this work the mean values and standard deviations of the remaining, quality-assured temperature and relative humidity measurements, assuming response times within 5 s as shown in Table 1.

### 2.3.2 Determination of amount fractions

In this work, we develop a correction method to accurately determine the amount fractions of NO and $NO_2$ using regression parameters determined with the ECS in-flight data (Sect. 3). A detailed description of the correction procedure is given in the supplement and is based on the method of Mead et al. (2013). After applying this correction method, the sensor voltage signals in mV are converted to amount fractions using sensitivity values in mV/ppb provided by the manufacturer (Table S1). Here, we assume constant sensitivities given that their dependency, e.g., to temperature, is a 2nd order effect (Mead et al., 2013;
Popoola et al., 2016).

To account for sensor response times, a low pass filter, in this case a centred moving average with a window size of 31 seconds is used that corresponds to the $t_{90}$, defined as the duration the sensor needs to reach 90 % of the final signal after a step change





in concentration. This value is derived from the combined information given by the manufacturer and by published measurements (Mead et al., 2013): the manufacturer provides a $t_{90}$ of < 45 s for NO and < 80 s for $NO_2$ from 0 to 2 ppm,

whereas Mead et al. provide a $t_{90}$ of 21 s for $NO_2$.

As a reference correction method, we use the recommended correction described by the manufacturer (Alphasense, 2019c) following the above equations to correct the NO and $NO_2$ *WE* output for effects of temperature:

$$WE_{NO,c} = (WE_u - WE_e) - k_T \times \left(\frac{WE_0}{AUX_0}\right) \times (AUX_u - AUX_e), \tag{1}$$

$$WE_{NO_2,c} = (WE_u - WE_e) - n_T \times (AUX_u - AUX_e), \tag{2}$$

where *WE* and *AUX* are the working and auxiliary electrode voltages. The subscripts u, e and 0 stand for the uncorrected, i.e., measured signal, the electronic offset, and the sensor zero, respectively. The electronic offsets and sensor zero values are provided by the manufacturer and given in Table S1. $WE_{NO,c}$ and $WE_{NO2,c}$ are the corrected working electrode voltages for NO and $NO_2$, respectively. The temperature compensation factors $k_T$ and $n_T$ are given in the range of -30 °C to 50 °C in 10 °C steps. For temperatures within these 10 °C steps, a linear interpolation is advised.

ECSs were also used to measure CO and Ox ($NO_2 + O_3$). A comparison of the ECS measurements to the MIRO reference instrument was performed after following the manufacturers recommendations to estimate amount fractions for CO and Ox. However, high uncertainties were found due to the high CO and $O_3$ backgrounds creating an offset to the sensors that was not accurately accounted for based on the manufacturer's correction procedure (Fig. S3). Additionally, the described correction procedure in the supplement used in this work for NO and $NO_2$ relies on periods with low analyte concentrations, ideally zero,

to account for offsets of the background signal. While this is a good approximation for NO and $NO_2$ it is not applicable to CO and $O_3$ because of their higher background concentrations often above several tens of ppb. Laboratory evaluation and optimization of the CO and Ox sensor performance is currently ongoing and the focus of a future study. In this work we evaluate the performance and highlight the potential of the NO and $NO_2$ sensors for accurate airborne applications.

## 3 Results and discussion

### 3.1 Sensor signal dependencies

All Zeppelin measurements were filtered for periods of low NO and $NO_2$ concentrations (< 2 ppb) using the measurement data of the reference instrument in order to find possible influences on the *WE* signal of the electrochemical sensors. This effectively reduces the voltage increase due to the analyte and leaves a signal that is primarily influenced by other factors. The correction method developed and described in the following is entirely independent of a reference device and requires only the sensors

used. Figure 3 and Figure S4 show the correlation of *WE* to *AUX*, *T*, and d*RH*/d*t* at different time resolutions (i.e., averaging intervals) for NO and $NO_2$, respectively. For 1 s resolution, coefficients of determination ($R^2$) are 0.87, 0.73, 0.82 for NO and 0.57, 0.13, 0.86 for $NO_2$, for *AUX*, *T*, and d*RH*/d*t*, respectively. As described in Sect. 2.1 *AUX* can be used to correct the *WE* signal from external interferences. The collinearity between *AUX* and *T,* that is shown by the colored dataset in Fig. 3 further

promotes the significant influence of temperature on the sensor measurements. However, other unknown interferences could have a simultaneous effect on *WE* and *AUX*, such as changes of the electrolyte composition or influences on the sensor boards electronics that control the voltages of the electrodes. Therefore, we also consider *AUX* as an additional correction parameter despite the observed collinearity with temperature.


**Figure 3: Scatterplots of NO sensor for *WE* vs. *AUX*, *T* and d*RH*/d*t* with different time resolutions of 1 s, 30 s and 120 s provided per row. The colors are used to show collinearity between *AUX* and *T*, and that the shift of WE voltages in the d*RH*/d*t* plot is a temperature interference.**



Figure 3 shows that temperature and *AUX* dependencies are effective on a longer timescale since the relationship with *WE*

does not change with lower time resolutions transitioning from 1 s to 30 s and 120 s. On the contrary, the correlation with d*RH*/d*t* decreases progressively with decreasing time resolution. Since ECS are mostly deployed for long-term monitoring of air quality at e.g., stationary monitoring stations, mean values of up to 1 hour are often used (Mijling et al., 2018). Therefore, at longer time resolutions such effects are filtered out. However, for mobile applications, a high time resolution results in a better spatial resolution of the pollutant distributions, making this effect relevant.

Previous publications have shown the influence of humidity changes on the *WE* signal, after which there is a spike in the signal followed by a slow decay (Mueller et al., 2017; Pang et al., 2018; Pang et al., 2017). Mueller et al. (2017) conducted laboratory tests with relative humidity changes of 5 % RH every 20 minutes between 40 % RH and 60 % RH. They observed that this changed the sensor signal by a similar order of magnitude as the addition of 70 ppb $NO_2$. Afterwards, the signal decreased exponentially in time back to equilibrium. Through further measurements, they found that the effect was dependent on the

magnitude and rate of the relative humidity variation that can affect the measurement accuracy over minutes to hours – but the physical reason for this is currently unknown. Pang et al. (2017) came to a similar conclusion. Rapid RH changes ($\approx$ 20 %/min) had an immediate influence on the sensor signal followed by a required recovery period of up to 40 min to restore the original value whereas small humidity changes of about 0.1 %/min had no significant effect. When compared to our measurements, the humidity changes in these laboratory experiments are unidirectional and over a longer time scale. During the flights, we

observe both negative and positive humidity changes on a short time scale. For example, the mentioned 20%/min *RH* change that triggers the long recovery phase results in 0.33 %/s, assuming a linear increase. During the Zeppelin flights we observe such a longer lasting effect starting at about 0.7 %/s, after which it takes up to 5 min, for the signal to stabilise again, depending on the magnitude of the rate of change. Since we do not have an exact analytical solution for correcting this dependency yet, these values were removed from the data set. However, we were able to describe the dependence of the *WE* voltages to the

temporally small-scale changes in relative humidity (max. $\pm$ 0.6 %/s), which affect the signals only immediately and briefly, with a linear relationship (see Fig. 3).

In the following, we use the above correlations on temperature, *AUX*, and d*RH*/d*t* to develop a correction method using the equations below for NO and $NO_2$.

$$WE_{NO,c}(t) = WE_{NO,a}(t) - \beta_0 \times \frac{dRH}{dt}(t) - (\alpha + \beta_1 \times \exp(\beta_2 \times T(t)) + \beta_3 \times AUX(t)), \qquad (3)$$

$$WE_{NO_2,c}(t) = WE_{NO_2,a}(t) - \beta_0 \times \frac{dRH}{dt}(t) - (\alpha + \beta_1 \times T(t) + \beta_2 \times AUX(t)), \qquad (4)$$

where $WE_a$ is the *WE* signal after the preparation steps from Sect. 2.3.1, *T* is the temperature in °C, and $\alpha$, $\beta_z$ (z = 0, 1, 2, 3) are the determined regression parameters. We give a more detailed description of how to obtain Eq. (3) and (4) in the supplement and show the regression parameters in Table S2. To calculate the amount fractions, $WE_{NO,c}$ and $WE_{NO2,c}$ are then divided by the corresponding sensitivities of the sensors (Table S1). In the following sections we will often use $NO_x$ (NO + $NO_2$). For the

ECS, this means





$$x_{NO_x}(t) = \frac{WE_{NO,c}(t)}{S_{NO}} + \frac{WE_{NO_2,c}(t)}{S_{NO_2}}, \tag{5}$$

in which the summands are moving averages as described in Sect. 2.3.2 to consider the response time of the ECSs.

## 3.2 Validation of ECSs performance

### 3.2.1 Intercomparison of ECS setups

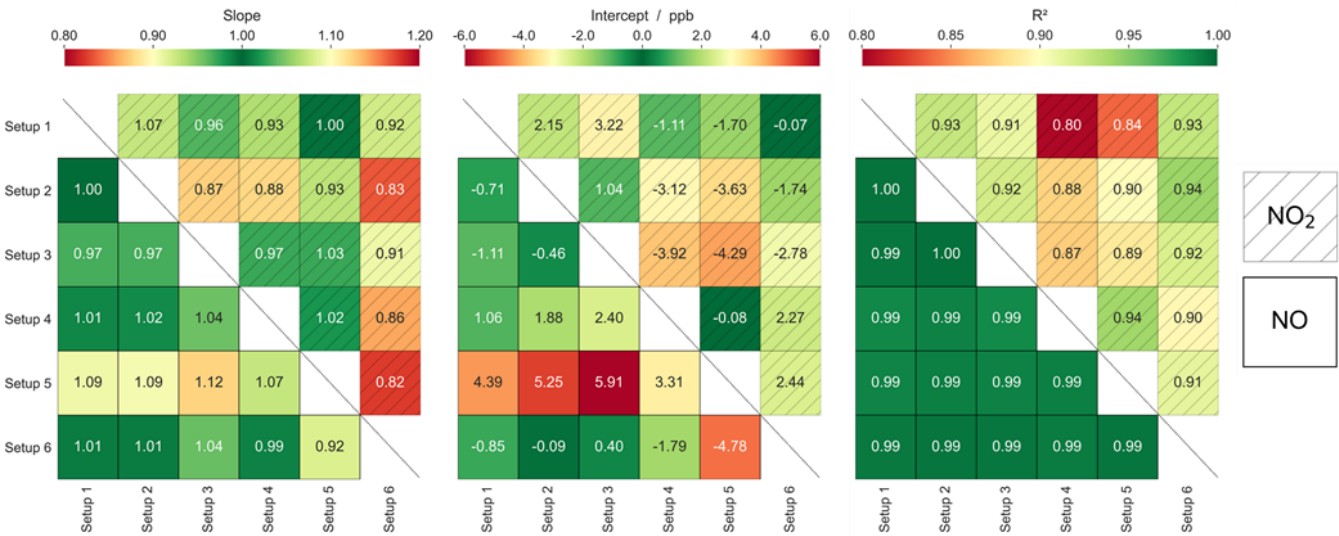

**Figure 4: From left to right: slopes, intercepts, and R² of the linear regressions between one setup and any other setup. The lower triangle, depicted with black framed rectangles are the NO sensors results. The grey diagonally striped rectangles in the upper triangle show the results of the NO₂ sensors. The red to green color map indicates the range from worst to best values, respectively.**

Figure 4 shows the intercomparison of all ECS setups after applying the corrections presented in Sect. 3.1, by performing linear regression analysis including their slopes, the intercepts, and coefficients of determination $R^2$. All NO sensors are in good agreement with slopes ranging from 0.92 to 1.12, intercepts from $\pm 0.09$ ppb to $\pm 5.91$ ppb and $R^2 > 0.99$, whereas for $NO_2$ slopes range from 0.82 to 1.07, intercepts from $\pm 0.07$ ppb to $\pm 4.29$ ppb, and $R^2$ from 0.80 to 0.94. Although the $R^2$ is high for all setups, the regressions of setups #4 and #5 with the NO sensors show greater variability in terms of intercepts and

slopes. For NO, setup #5 has the highest offset of 5.91 ppb and a larger slope compared to all other setups. For $NO_2$ the results are not as definite but the $R^2$ is generally lower for setups #4 and #5, indicating higher noise of the sensors. This is also further supported by Fig. S5 with the noise of the sensor setups #4 and #5 at 5.03 ppb and 3.59 ppb for NO, and 6.32 ppb and 5.43 ppb for $NO_2$, respectively. Setups #3 and #6 have similar precisions for NO compared to setup #2 with values of 1.45 ppb and 1.98 ppb compared to 1.95 ppb, respectively; however, their noise for $NO_2$ is approximately 55 % and 61 % higher than for

setup #2. Therefore, we exclusively use setup #2 in the following given the consistent agreement to other sensors, the lower noise, and the high in-flight data coverage > 98 %.





The regression parameters for $NO_2$ are more variable than for NO. One possible reason is the smaller temperature dependence for $NO_2$. As a result, *AUX*, for example for setup #2, varies only between 220 to 240 mV, whereas NO varies between 300 to 400 mV (Fig. 3 and S4). This indicates that the temperature influence is the dominant contribution for the baseline correction

of the NO sensor, whereas the *WE* of the $NO_2$ sensor could additionally be influenced by other interfering factors that do not affect the *AUX* and are not accounted for here. Another factor is the much larger number of data points for NO (n = 4026) than for $NO_2$ (n = 155) for amount fractions above 50 ppb, leading to a more stable regression line for NO.

### 3.2.2 Noise reduction

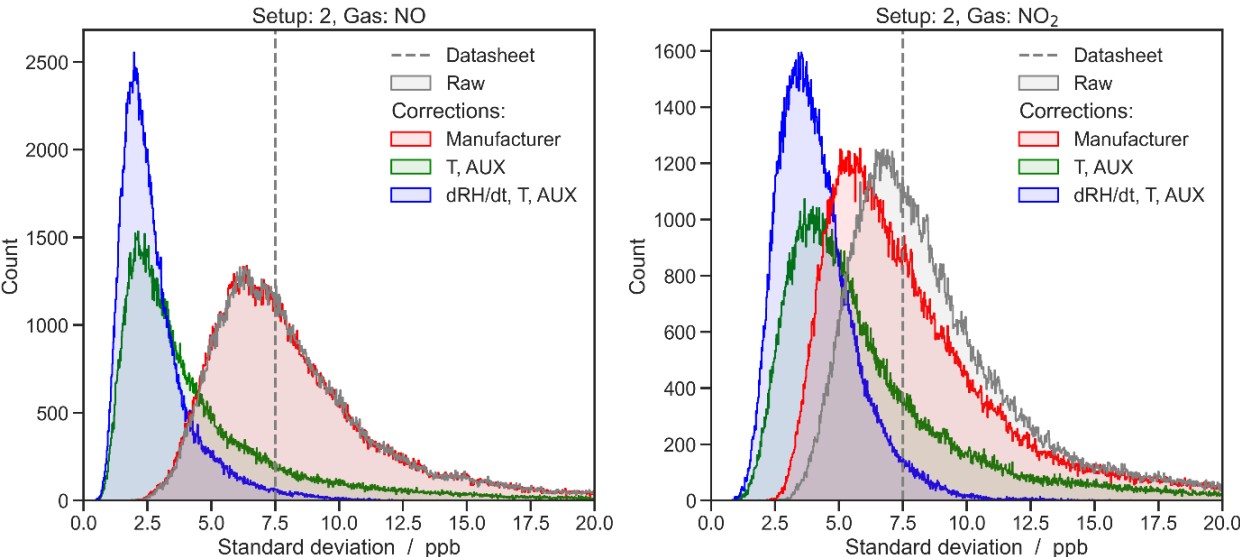

**Figure 5: Distributions of standard deviations calculated inside 31 s windows. Adding d*RH*/d*t* to the correction procedure leads to a narrower distribution by reducing high changes of the signal on a short timescale.**

Figure 5 shows the distributions of the standard deviations of the NO and $NO_2$ sensors for each 31 s windows which were used to calculate the moving average (Sect. 2.3.2). The data was filtered for periods with height above 100 m (n = 270 198) to avoid engine exhaust peaks that would result in high standard deviations because of the high changes in concentrations inside these

time windows. The correction used here, shown in blue and green, results in the lowest noise with modal values for NO of 2.03 ppb for the correction with *T* and *AUX*, and 1.95 ppb when d*RH*/d*t* is additionally included in the correction, whereas for $NO_2$, the noise modal values are 3.66 ppb and 3.32 ppb. Besides the change in peak position, the distributions become narrower when correcting with d*RH*/d*t*. This leads to significantly higher peaks of around 60 % for both analytes because the number of data points with standard deviations above 3 ppb for NO and above 5 ppb for $NO_2$ decreases, highlighting the significant noise

reduction when including d*RH*/d*t* in the correction procedure.





### 3.2.3 Comparison of ECSs with the MIRO MGA

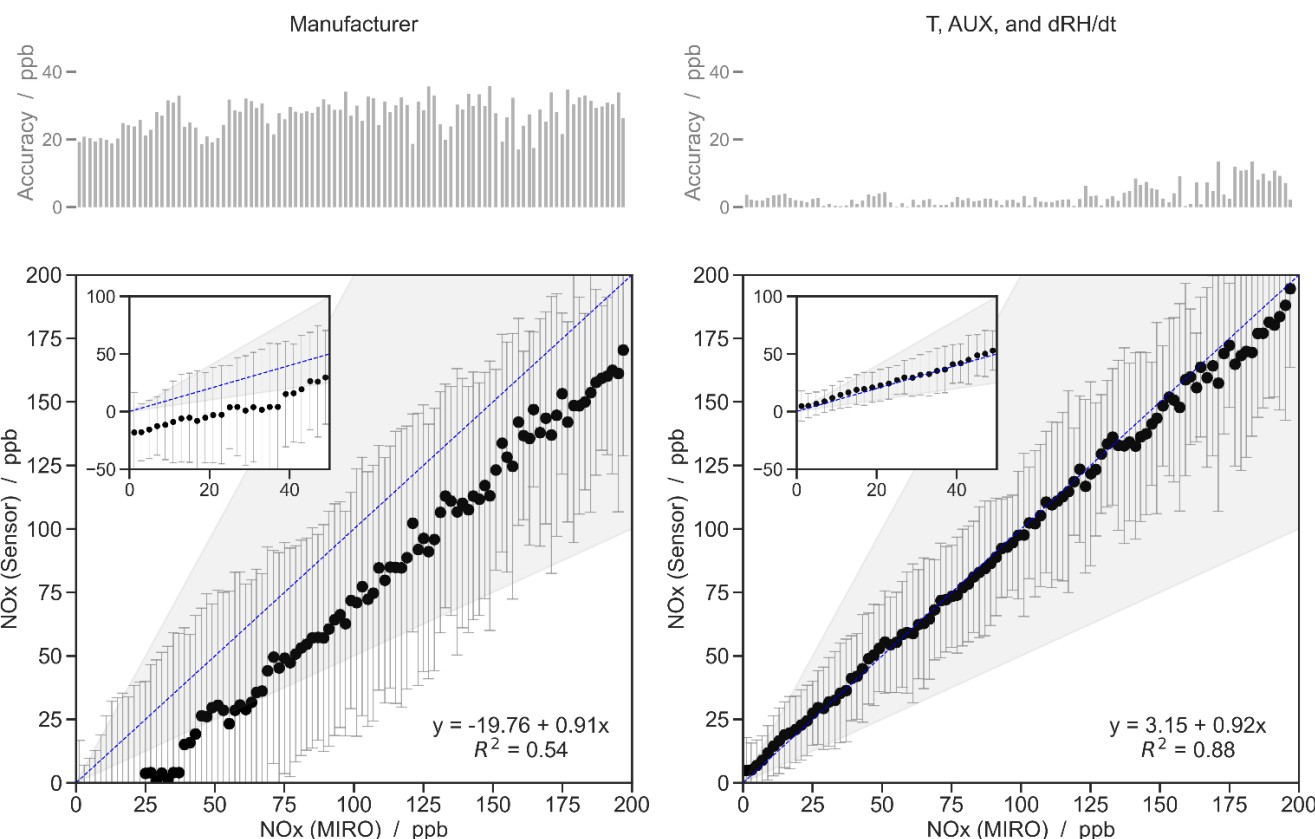

**Figure 6: Scatterplots of the corrected NO$_x$ sensor data vs. MIRO MGA (reference) data classified in 2 ppb bins and two times the standard deviations ($\pm\,2\sigma$). On top the accuracies for each bin are shown. Corrections as recommended by the manufacturer on the left, our correction method on the right. The dashed blue line marks the 1:1 line and the range of 2:1 and 1:2 is shaded in grey. The inlet plots show the data in a higher resolution from 0 to 50 ppb of the reference instrument. In addition, the linear regression results are shown on the bottom right.**

Figure 6 shows the amount fractions of NO$_x$ following the corrections recommended by the manufacturer as described in Sect. 2.3.2 and the correction method developed in this work compared to the MIRO MGA used as a reference device. As shown in the upper plots the deviations of the sensor values to MIRO decreased from an absolute average of $27.3 \pm 4.8$ ppb to $3.5 \pm 3.1$ ppb. This means an absolute accuracy improvement by nearly an order of magnitude. The accuracy increase is mainly the result of the improved offsets changing from -13.15 ppb, -8.29 ppb, and -19.76 ppb to 9.12 ppb, -6.76 ppb, and 3.15 ppb for NO, NO$_2$ (Fig. S6), and NO$_x$, respectively. Precision also improved, reflected by the decrease of the associated error bars in Fig. 6 resulting in a higher coefficient of determination from 0.54 to 0.88. In general, NO and NO$_2$ measurements corrected with our method, and the resulting NO$_x$ values are close to the 1:1 line. Moreover, the two standard deviations corresponding to 95 % of the data in each bin, are within a factor of 2 above 20 ppb, which is particularly driven by the d$RH$/d$t$ correction, as also shown in Fig. 5.





While on average there is agreement within 3.5 ppb for $NO_x$ between the MIRO and ECSs at both high and low concentrations this agreement is not evident for the lower amount fractions of NO or $NO_2$ (Fig. S6). For NO an average overestimation of

34.4 % is observed below 40 ppb, that increases to an average deviation of up to 600 % in the range of 0 to 5 ppb, whereas for $NO_2$ an underestimation of 31.3 % is observed for amount fractions below 25 ppb, increasing to 300 % below 5 ppb, because of the small absolute numbers. It is possible that the 8 m sampling line to the MIRO could influence the composition of NO and $NO_2$: as an in-line reaction of NO with $O_3$ could lead to higher $NO_2$ and lower NO concentrations, compared to the sensors that have no sampling line. However, this effect is expected to be minor due to the short sample residence time of 5 s. In

addition, other parameters may influence the performance of the electrochemical sensors compared to the MIRO, such as the wind speed (Mead et al., 2013) or the atmospheric pressure. To quantify these possible influences, a measuring chamber together with a pump will be used in future campaigns to provide a constant volumetric flow. Additionally, pressure sensors will be installed on the measurement boards in future.

Evidently, with the manufacturer's correction, amount fractions in the low ppb range cannot be quantified (Fig. 6)

predominantly due to the high offset of -19.76 ppb. One possible reason is that in Eq. (1) and (2) the same temperature compensation factors are used for all sensors, whereas each sensor may react differently to a temperature change, which is also shown by our regression parameters in Table S2. Another cause could be a drift of the zero value, which is not caught by the temperature and *AUX* correction in these equations. In our method, sensor drifts are corrected by using minimum values of measured *WE* voltages following the procedure described in Mead et al. (2013) as discussed in Sect. 2.3 and the supplement.



## 3.3 Detection of anthropogenic NOₓ emission sources

**3.3 Detection of anthropogenic NO$_x$ emission sources**

**Figure 7: Amount fractions of NOx are shown by color and size for the MIRO MGA (a, c) and the corrected sensor data (b, d) during flights in North Rhine-Westphalia, Germany, near and in the cities of Eschweiler and Duisburg, respectively (© OpenStreetMap contributors 2022. Distributed under the Open Data Commons Open Database License (ODbL) v1.0). The arrows orientation and length indicate the wind direction and wind speed (2.1 to 7.2 m s$^{-1}$ for (a, b) and 1.0 to 2.6 m s$^{-1}$ for (c, d)), respectively. With this, the emission sources can be narrowed down to a lignite-fired power plant located southeast of the detected peak (a, b) and to a steel industry (c, d), located in the grey shaded area.**





After the successful validity check by comparing the corrected sensor data with the reference device in the former section, we
now present possible applications and the potential of ECS for airborne measurements. Figure 7 (a) and (b) show the flight
path on 06 May 2020 between 07:30 and 08:30 UTC in Eschweiler, North Rhine-Westphalia, Germany when using the
reference instrument and the ECS sensors, respectively. Another example is shown in Fig. 7 (c) and (d) during a Zeppelin
flight over Duisburg Nord (Hamborn), North Rhine-Westphalia, Germany on 07 May 2020 between 13:30 and 14:15 UTC.
As depicted in Fig. 7 (a) $NO_x$ background concentrations are $4.9 \pm 2.0$ ppb and increase significantly in the northwest to a
maximum of $21.3 \pm 2.2$ ppb due to the emissions from the lignite-fired power plant located near the flight path (Tillmann et
al., 2022), further supported by the wind data that were extracted from high-resolution model simulations. When applying the
correction method recommended by the manufacturer for the ECSs (Sect. 2.3.2), background concentrations of $NO_x$ are at -
$8.6 \pm 5.2$ ppb and the industrial plume emissions at $11.3 \pm 17.3$ ppb highlighting the limitations of this method. However, after
applying the correction method developed in this work, shown in Fig. 7 (b), a significant improvement is observed with
background concentrations at $3.5 \pm 4.2$ ppb and the power plant plume average at $21.4 \pm 6.6$ ppb (Fig. S7 (a)). This results in
40 % higher background concentrations for the sensor measurements in comparison to the MIRO and 5 % lower concentrations
for the in-plume measurements.

Similar results can be seen in Fig. (c) and (d); Duisburg is known for its industry – there is a steel mill in this district, which is
supplied with electricity by the gas-fired combined heat and power plant located on the site. In addition, the motorways A42
and A59 run through the area. Westerly winds indicate that the source of emissions during the observed maximum amount
fractions of $37.7 \pm 6.1$ ppb for the MIRO and $47.1 \pm 6.4$ ppb (Fig. S7 (b)) for the sensor measurements, are from the steel mill
and the heat power plant. The influence of highway emissions on the $NO_x$ concentrations is observed at around 51.48° N and
6.77° E when Southerly wind directions transported emissions from the A42 and A59 highways to our sampling line. This is
especially evident for the sensor data (d) but also observable for the MIRO (c). Here, the peak values are $13.8 \pm 0.7$ ppb and
$22.6 \pm 3.6$ ppb for the MIRO and the ECS, respectively.

These case examples, highlight the potential of the ECS to detect emission sources with concentrations down to 20 ppb.
Although, there are larger relative uncertainties due to the deviations of the individual NO and $NO_2$ sensors (see Sect. 3) the
ECSs show better agreement with the MIRO at higher concentrations, especially for $NO_x$ further promoting their potential for
airborne applications.

## 4 Conclusions

We showed that electrochemical sensors (ECS) can be successfully used for airborne in situ measurements in the PBL. For
this, we used a Zeppelin NT to perform an in-flight comparison of six sensor setups with a reference device during two
measurement campaigns, including targeted and commercial flights, in Germany from April to June 2020. Each setup consisted
of four electrochemical sensors for CO, NO, $NO_2$, and Ox ($NO_2 + O_3$) and one sensor for temperature and relative humidity.



These were installed in a custom-built hatch box mounted under the gondola. A quantum cascade laser-based multi-compound gas analyzer, called MIRO MGA, was placed as a reference instrument inside the gondola with an inlet line beside the ECSs. We developed a stand-alone correction method, i.e., independent of a reference device, for the ECSs that accounts for external influences on the NO and $NO_2$ sensor signals by using the variability of the auxiliary electrode voltage, temperature, and the relative humidity gradient. We show that this correction method substantially improves the accuracy down to $6.3 \pm 5.7$ ppb

and $8.7 \pm 5.9$ ppb and lowers the noise of the ECS sensors to 1.92 ppb and 3.32 ppb for NO and $NO_2$, respectively. The combination of both sensor types (NO and $NO_2$) leads to a further improved accuracy of $3.5 \pm 3.1$ ppb for $NO_x$. When compared to the MIRO MGA good agreement with a coefficient of determination of 0.88 and a slope of 0.92 is achieved for $NO_x$ measurements. However, at lower concentrations below 40 ppb and 25 ppb average deviations of 34.4 % and -31.3 % for NO and $NO_2$, respectively, were evident. Below 5 ppb, these deviations increased up to 300 % and -600 %, because of the

small absolute values, indicating the limitations of ECSs for accurate quantification at lower amount fractions.

We highlight the potential to use the sensors for emission source identification during the Zeppelin flights by identifying emissions from a lignite-fired power plant with a peak of approximately 21 ppb of $NO_x$ and from a large industrial area in Duisburg with a peak above 40 ppb.

Results from this work emphasize the potential of these sensors for in situ airborne applications and provide a first milestone

for future quality-assured use onboard UAVs without the need of a reference device. A comprehensive characterization in the laboratory, including the simulation of airborne conditions, before and after such applications, will improve the ECS data quality even further.

*Data availability.* The data are available at https://doi.org/10.26165/JUELICH-DATA/6D8B70 (Schuldt et al., 2022).


*Author contribution.* RT, FR, and AKS designed the experiments and flight campaigns. BW, CW, and FR carried them out. GIG and TS visualized the data. AKS, TAJK, GIG, RT, and TS contributed to the interpretation of the results. GIG and TS prepared the manuscript with contributions from all co-authors.

*Acknowledgments.* We acknowledge the support of Deutsche Zeppelin Reederei (DZR) and Zeppelin Luftschifftechnik GmbH (ZLT). We would like to thank Anne Caroline Lange, Elmar Friese and Philipp Franke for the high-resolution WRF and EURAD-IM simulations, and Morten Hundt and Oleg Aseev for instrumentation support of the MIRO MGA. The authors gratefully acknowledge the computing time granted through JARA on the supercomputer JURECA at Forschungszentrum Jülich.


The authors declare that they have no conflict of interest.



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
