# Peer review of "Electrochemical sensors onboard a Zeppelin NT: In-flight evaluation of low-cost trace gas measurements"

_Atmospheric Measurement Techniques, 2022_

## Author Comment (AC1)

**Response to Reviewers**

Electrochemical sensors onboard a Zeppelin NT: In-flight evaluation of lowcost trace gas measurements

\_\_\_\_\_

Referee comments are in **black** and authors responses are in **blue**

**Reviewer #1:**

Summary

"Electrochemical sensors onboard a Zeppelin NT: In-flight evaluation of low-cost trace gas measurements" evaluates the performance of a suite of sensors installed in a hatch box within the bottom of an airborne platform. The focus here is on NO and NO2 measured by Alphasense electrochemical sensors. Six units are flown together underneath the Zeppelin and intercompared. Other aspects, including results from a reference mid-infrared MIRO instrument, were described in a related paper by the authors. Many papers have been published in recent years on low-cost gas and particulate matter sensing, along with their calibration and correction for spurious environmental dependencies. In my opinion, the manuscript is well written and scientifically sound, although the scope is fairly limited. The most novel aspect is airborne deployment in Zeppelin flights, which took place in Germany.

We thank the reviewer for the positive feedback and the helpful comments. Response to each comment is provided below.

**Main**

My main comment is that it feels like the reference MIRO data from the flights is underutilized in terms of validating the performance of the electrochemical sensors, especially since laboratory test data is not included. Some thoughts on this point:

- While the intercomparison of the six setups in Figure 4 is interesting, is the conclusion about setup #2 performing best also supported by comparing against MIRO MGA? What do those results look like?

This comparison is shown in Figure 6. We now added more information to the caption of Figure 6 and a reference in the text to clarify that these plots show the comparison of setup #2 with the MIRO MGA. Moreover, we now added Figure S10 in the supplement that shows the linear regression of each sensor relative to the MIRO MGA. These results highlight the consistent agreement of all sensors to the MIRO MGA with the slopes of the linear fits for NOx agreeing within 10 %. We now clarify in the text why we choose setup #2 for further evaluation of the sensors as follows:

"The clocks on each of the six sensor setups were manually pre-set, therefore time synchronization was not ensured. In the first step we chose a *master setup*, here setup #2 that was operational throughout all flights with a data coverage of 99 %."

- Corrections were derived to remove dependences on T, AUX, and dRH/dt without using MIRO, and compared against manufacturer recommended corrections. This is advantageous in avoiding requiring use of a reference instrument. However, can MIRO be used independently here to evaluate how well this correction approach works?

Figure 6 (d) for setup #2 and Figure S10 for all setups now show the linear regressions of the corrected data vs. MIRO. The  $R^2$  range from 0.75 to 0.88 which reflects the good performance of the correction method. We further clarify this in the main text as mentioned in the comment above.

- What can be said about stability of the ECS sensors and derived calibration during or between flights.

The measurements performed in this paper range from end of April to mid-June. When accounting for dependencies to T, AUX, dRH/dt the correlation of the ECS sensors to the MIRO reference system are consistent highlighting that no major instabilities were present during these measurements (Fig. 6 and S10). Wei et al. (2018) highlighted that for a 2 month period, like in our study, ECS sensors were stable with drifts < 2 ppb/month. We added a few sentences and references in the main text (Sect. 3.2.3) highlighting the importance of long term measurements to further evaluate long term instabilities.

"Besides the above direct influences, there is also the possibility of sensor drifts, i.e., a change of the sensor signal with time. Wei et al. (2018) estimated a possible drift of < 2 ppb/month whereas Mead et al. (2013) state that the sensitivity of the sensors remained unchanged over an 11-month measurement period. For our deployment duration of 1.5 months, sensor drifts are therefore expected to be within the uncertainty of the measurements which is also reflected by the good agreement of the ECS and the MIRO in Fig. 6 and S10. Furthermore, Fig. S11 shows the timeseries of all sensors during different flight days in May and June to evaluate the influence of such sensor drifts. The consistent correlation of all setups to the MIRO highlights the stability of the sensors during this study. However, we promote the need for controlled laboratory measurements in the future to evaluate long-term influences on the stability of the ECS signals including sensor drifts."

**Minor**

I am not completely sure what data was used to generate the figures. On L265 it talks about only showing setup #2. L160 talks about excluding #4, #6, and partly #5. Which sensor setups are included in Fig 3, 5-7?

We split up section 2.3.1 and added a new section 2.3.2 to clarify that only the temperature and relative humidity sensors of setups #4, #5 and #6 were excluded. We also added the setup number in all corresponding captions for clarification.

Similarly, Figures 4 and 6 seem to be showing aggregate data over all the flights. This could be clearer. How many individual flights and hours of data are included? Is any data excluded?

We added more information to the captions of Fig. 4-6. This includes the hours of measurements ( $\approx 286$  hours total,  $\approx 75$  hours in-flight), the measurement height, and the number of data points. Additionally, we extended Figure 6 by adding a colorbar that shows the number of flight measurements performed per bin.

Figure 1(a) what is the height and width of the hatch box?

We added the following in section 2.1: "The hatch box dimensions are  $738 \times 538 \times 162$  (length  $\times$  width  $\times$  height in mm)."

L160 "From this correlation analysis it is evident that the sensors of setups #4, #6 and partly #5 provide erroneous data."This is not obvious to me. Figure S2 in particular is referenced in the sentence before, but is hard to read both in terms of the font size within the figure and having a fairly brief figure caption.

We agree and thank the reviewer for pointing that out. We now replaced the figure, improved the caption, and added section 2.3.2 to further describe the selection process. Fig. S4 now shows the differences between each temperature sensors to the rest and highlights that sensors #4, #5, and #6 deviate more from the others.

Figure 6; is top panel with y-axis label 'Accuracy / ppb' showing the +/- 2 standard deviations as shown in the bottom panel? This should be clearer.

The standard deviations  $\pm 2 \sigma$  reflect the precisions of the measurements. We added the following to the caption to better define the accuracy: ", i.e., the absolute discrepancies between sensor and MIRO MGA data".

"Evidently, with the manufacturer's correction, amount fractions in the low ppb range cannot be quantified (Fig. 6) predominantly due to the high offset of -19.76 ppb."

I don't understand this point since an offset affects low ppb as well as high ppb measurements equally and does not determine the sensitivity of the measurement. It could be an issue if readings are filtered to be above zero.

That is a valid point. Rephrased the sentence to "Evidently, with the manufacturer's correction, amount fractions (Fig. 6(a)) cannot be accurately quantified predominantly due to the high offset of -19.76 ppb."

Acknowledgements: WRF is not directly mentioned in the main text, although perhaps it was run to generate the wind field in Figure 7. EURAD-IM is not mentioned.

We added "EURAD-IM, WRF" to the caption of Fig. 7.

---

## Author Comment (AC2)

**Response to Reviewers**

**Electrochemical sensors onboard a Zeppelin NT: In-flight evaluation of lowcost trace gas measurements**

\_\_\_\_\_

Referee comments are in **black** and authors responses are in **blue**

**Reviewer #2:**

I think it is a really good platform for low-cost sensors to measure the boundary layer. Please find some comments and questions below about the manuscript and methods that were used.

We thank the reviewer for the positive feedback. Response to each comment is provided below.

**Time synchronization**

What is LTE?

We added the abbreviation in section 2.1 as follows: "Long Term Evolution (LTE) – a standard for wireless broadband communication for mobile devices – equipment for remote access, [...]".

Did you apply the Savitzky-Golay filter before or after you had applied your temperature correction? How noisy were the raw data versus after this filter had been applied. Did the analyte peaks stand out prominently from the noise?

We applied the Savitzky-Golay filter before the temperature correction. We now added two new figures in the supplement to address your comments. Fig. S2 shows the noise distributions of the background signals before and after applying the Savitzky-Golay filter. Fig. S3 indicates the range of values that is reduced. For that, we calculated the differences of the sensor signals before and after applying the filter. The reduction is within 10 ppb (2  $\sigma$ , corresponding to around 95 % of the data), i.e. similar to the noise levels shown in Fig. S2. Therefore, larger peaks above the detection limit of the sensors, for example as shown in Fig. 7, are not affected by the filter. This is now also highlighted in Sect. 2.3.1:

"Fig. S3 shows the difference between the sensor signal before and after applying the filter at different NO and NO2 concentrations. The signal reduction is within 10 ppb (2  $\sigma$ , corresponding to around 95 % of the data), i.e., similar to the noise levels of the sensors (Fig. S2), independent of the NO and NO2 concentrations, which highlights the minor influence of the Savitzky-Golay filter at larger peaks."

**Determination of amount fractions**

Not quite sure how you reached a window size of 31 seconds for the t90 value. Did you run the EC sensors in the laboratory under controlled conditions and then change the [NO] and [NO2] to concentrations that you'd expect to see on the Zepplin and calculate it from there?

Was Mead et al 2013 also using these sensors on a Zepplin or aircraft? Was the concentration range in that paper the same as you'd expect for your deployment?

You use the fact that Mead et al 2013 found a t90 value of 21 sec for NO2. What was Mead et al t90 value for NO? The manufacturer provides vastly different t90 for NO2 and NO. Also do we expect that the EC sensor technology hasn't changed in ten years so you and Mead et al have the same versions of EC?

That is a great point. We now added our laboratory results in Table S2 that show  $t_{90}$  of 34.6 s  $\pm$  2.0 s and 28.1 s  $\pm$  1.6 s for NO and NO2, respectively, in agreement with the manufacturer and published measurements. The following is also added in Sect. 2.3.2:

"This value is derived from the combined information given by the laboratory measurements (Table S2), in agreement with the manufacturer and by published measurements (Mead et al., 2013): [...]"

**Sensor signal dependances**

So you assume that an increase in NO or NO2 from 0 ppb to 2 ppb does not cause any increase of EC. Did you check this in the lab?

The limit of detection (LOD) defined as as three times the standard deviation of the baseline  $(3\times\sigma)$  is 6.0 ppb and 10.5 ppb for NO and NO2, respectively. Therefore, 2 ppb is below the detection limit of the ECS and we consider it a valid approximation here.

Why did you not find the median NO signal (of the six NO EC) and subtract that from each NO sensor to get the 'signal primarily influenced by other factors'? I think that would be more appropriate rather than assuming a step change of the analyte concentration doesn't impact the EC output at all.

We now rephrase this sentence to better clarify the steps followed here: "This procedure reduces the data to only background concentration periods and provides the signal that is influenced by cross interferences.".

Why did you correlate WE to the change in humidity and not the humidity? Or if you wanted to use the differential, why not use that for temperature also, dT/dt? Changing the way these are plotted makes it difficult to see quickly the difference between the temperature and the humidity dependance.

We added Fig. S7 in the supplement and highlight that we could not detect any impact from dT/dt in Sect. 3.1.

**Validation of ECSs performance**

I think there needs to be a timeseries of the 6 NO/NO2 sensors showing how they all behaved over the duration of the flights. The slopes look good but did they start out correlated and then drift apart as they all respond differently to each interfering variable? Or were they all slightly offset and this stayed consistent throughout? Maybe add temperature and humidity onto this too, with a shared time axis.

We added multiple time series of amount fractions, T and RH for different dates to the supplement (Fig. S11) and a discussion about the stability in the main text (Sect. 3.2.3). We found a constant offset for setup #5 that was also shown in Fig. S10.

**Comparison of ECSs with the MIRO MGA**

Accuracy plots are nice but please add the x-axis. Is this over time? In which case the accuracy decreases with the T,AUX and dRH/dt correction towards the end of the experiment. Or is it a shared axis with the

plot below – so accuracy decreases at higher NOx (MIRO) concentrations? That seems a little odd. Why would the accuracy decrease if the reference instrument is measuring more than 125 ppb NOx? I think if this is the case a little description of why that might happen is needed.

It is a shared x-axis. We now added the spline and ticks for the accuracy plot axis and colored the scatterplot by the number of flight measurements performed per bin. Because of the low number of data points above 125 ppb the variability is relatively high which is reflected by the precision (error bars) and also leads to the lower accuracies observed.

Both the EC NO and NO2 concentrations are overestimated (even at the higher concentrations you mention a 30 % overestimation). Was this a consistent overestimation throughout? Did you zero the EC amount fractions to the MIRO at the beginning of the experiment? It would be interesting to see if this overestimation changed over a 20 hour flight or over the entire campaign. I was under the impression the EC drift apart over a matter of hours – days.

In Sect. 3.2.3, we state that  $NO_2$  was underestimated for measurements below 25 ppb and NO overestimated for measurements below 40 ppb (see Fig. S9). We did not zero the amount fractions to the MIRO at the beginning of the experiment, because we developed a correction method that is independent of a reference device. As mentioned in one of the previous comments we added a short discussion about ECS stability in section 3.2.3 and multiple time series for different flight days in Fig. S11. We now reference Wei et al. (2018) in the main text who found a drift below 2 ppb/month.

I know you mention that the AUX electrode is supposed to remove all impacts of variables other than temperature but I think you could write a few sentences about the NOx EC response to other gases. Did you look at how the NOx EC responded to cross interferences? Was the signal influenced by any of the other compounds measured by the MIRO? i.e. did changes in the ozone concentration cause a response in NOx EC?

We added a Figure in the supplement (Fig. S12) to evaluate a possible cross interference of the corrected sensor signals to the other compounds measured by the reference device. We could not find any significant cross interference for the atmospheric concentrations that were present. This is now also highlighted in section 3.2.3:

"In this study, measurements of gases are limited to CO, CO2, CH4, N2O, H2O, and O3 performed by the MIRO MGA (Tillmann et al., 2022). No significant cross interference was observed under the present atmospheric concentrations, as shown in Fig. S12."

**Conclusions**

Deviations of the sensor values to MIRO decreased from an average of 27.3 to 3.5 ppb. What is the accuracy of the MIRO? The concentrations of NOx observed by this reference instrument seem to be pretty low; the Weisweiler timeseries shows that for this flight the majority of the NOx was around 5 ppb. A deviation of 3.5 ppb is quite a lot for concentrations in this range.

The precision of the MIRO MGA provided by the company is 0.3 ppb for NO and 0.08 ppb for NO2. We agree that 3.5 ppb is high in the mentioned background concentration range below 5 ppb. We highlight limitations of the sensors in section 3.3: "These case examples, highlight the potential of the ECS to detect emission sources with concentrations down to 20 ppb. Although, there are larger relative uncertainties due

to the deviations of the individual NO and  $NO_2$  sensors (see Sect. 3) the ECSs show better agreement with the MIRO at higher concentrations"

I think you successfully show that you can improve the accuracy of the NO NO2 EC sensors by correcting for temperature and humidity but I am not sure you could reliably deploy EC sensors in the Zepplin with no reference device and trust the output. It does seem to qualitatively pick up the NOx emissions from the power plant and industrial regions but there is still large amounts of uncertainty (below 5 ppb deviations between MIRO and EC increased up to 300 % and -600 %; the MIRO reported ambient NOx mixing ratios of 5 ppb for the majority of the time).

We agree with the reviewer that low-cost sensors aboard the Zeppelin can only be used for the detection of significant emission plumes. We further highlight this by adding a colorscale in Figure 6 that shows that the majority of ambient NOx concentrations is in the range below 20 ppb and further discuss it in the conclusions:

"However, at lower concentrations below 40 ppb and 25 ppb average deviations of 34.4 % and -31.3 % for NO and NO2, respectively, were evident. Below 5 ppb, these deviations increased up to 300 % and -600 %, because of the small absolute values, indicating the limitations of ECSs for accurate quantification at lower amount fractions."

On page 11, you also say that you only ended up using sesor set up #2 for the noise plots and intercomparison with MIRO. Really, this means that 1 out of 6 EC sensors can be used on a UAV platform without a reference instrument. Or would you still advise having 6 NOx EC and then taking the best performing one?

We now include Figure S10 to highlight that the majority of EC sensors showed agreement within 10 % with the reference instrument. The main reason we chose setup #2 for this analysis was the high data coverage during all of the flights that we now highlight in section 2.3.1:

"The clocks on each of the six sensor setups were manually pre-set, therefore time synchronization was not ensured. In the first step we chose a master setup, here setup #2 that was operational throughout all flights with a data coverage of 99 %."